# Deep Learning-Based Automated Diagnosis for Coronary Artery Disease Using SPECT-MPI Images

**DOI:** 10.3390/jcm11133918

**Published:** 2022-07-05

**Authors:** Nikolaos I. Papandrianos, Anna Feleki, Elpiniki I. Papageorgiou, Chiara Martini

**Affiliations:** 1Department of Energy Systems, University of Thessaly, Gaiopolis Campus, 41500 Larisa, Greece; annafele1@uth.gr (A.F.); elpinikipapageorgiou@uth.gr (E.I.P.); 2Department of Diagnostic, Parma University Hospital, Via Gramsci 14, 43125 Parma, Italy; chiaramartini10@gmail.com; 3Department of Medicine and Surgery, Section of Radiology, University of Parma, Maggiore Hospital, Via Gramsci 14, 43125 Parma, Italy

**Keywords:** cardiovascular diagnosis, coronary artery, deep learning, SPECT, myocardial perfusion images, classification

## Abstract

(1) Background: Single-photon emission computed tomography (SPECT) myocardial perfusion imaging (MPI) is a long-established estimation methodology for medical diagnosis using image classification illustrating conditions in coronary artery disease. For these procedures, convolutional neural networks have proven to be very beneficial in achieving near-optimal accuracy for the automatic classification of SPECT images. (2) Methods: This research addresses the supervised learning-based ideal observer image classification utilizing an RGB-CNN model in heart images to diagnose CAD. For comparison purposes, we employ VGG-16 and DenseNet-121 pre-trained networks that are indulged in an image dataset representing stress and rest mode heart states acquired by SPECT. In experimentally evaluating the method, we explore a wide repertoire of deep learning network setups in conjunction with various robust evaluation and exploitation metrics. Additionally, to overcome the image dataset cardinality restrictions, we take advantage of the data augmentation technique expanding the set into an adequate number. Further evaluation of the model was performed via 10-fold cross-validation to ensure our model’s reliability. (3) Results: The proposed RGB-CNN model achieved an accuracy of 91.86%, while VGG-16 and DenseNet-121 reached 88.54% and 86.11%, respectively. (4) Conclusions: The abovementioned experiments verify that the newly developed deep learning models may be of great assistance in nuclear medicine and clinical decision-making.

## 1. Introduction

Recent evidence suggests that coronary artery disease (CAD) is the most common type of cardiovascular disease being the leading cause of mortality worldwide. The percentage of death attributed to CAD is higher compared with other heart diseases. CAD occurs when the blood vessels are narrowed, and blood does not circulate properly around the heart. It is extremely crucial for a patient to be diagnosed early with any form of heart disease, especially CAD, so that the doctor can ensure that the patient will receive the proper medication and treatment [1,2].

A well-established method in clinical practice for the assessment of CAD is the single-photon emission computed tomography (SPECT) myocardial perfusion imaging (MPI) with technetium-99 m-labeled (99 mTc) perfusion agents. The method is rather advantageous for most cases. This method can offer rest and stress representation of the patient’s heart to identify areas that have myocardial perfusion abnormalities [3]. The most important factor is that SPECT offers three-dimensional information, as well as reduces scanning time, and decreases the procedure’s cost [4,5]. The SPECT clinical diagnostics methodology is the most popular technique in dealing with CAD, proving that computer-aided medical diagnostics to be increasingly used in recent years and offering doctors time saving and error avoidance. However, the increasing accuracy of computer-aided medical diagnostics is not only due to the advancements made in computer tomography and the newly developed image capture hardware that developed, but also due to the research development and innovation achieved in the deep learning (DL) and other machine learning (ML) methodologies.

DL-based solutions are increasingly being used to solve medical image analysis and computer-assisted intervention problems in medical informatics. This is due to their ability to effectively extract patterns and features only from input images, achieving tremendous results. The most significant advantage of such methods lies in the fact that they are adaptable to the specific functionality of medical image analysis. This advantage has been extensively utilized by researchers in the field in adapting DL into a variety of medical diagnostics with minimal implementation and heterogeneous infrastructure harmonization efforts. Additionally, DL provides a pipeline for medical imaging applications such as segmentation, regression, image generation, and representation learning in medical diagnostics [3]. DL is an advancement of artificial neural networks (ANNs) that consists of more layers that allow for higher levels of abstraction and better data predictions. Thus, the method currently has emerged to be the most powerful ML tool in the general imaging and computer vision domains and therefore in the medical informatics domain.

CAD systems are increasingly utilized in scientific diagnostics and they have an irreplaceable role in the diagnosis of MPI images, as they represent the blood flow of the heart in high contrast [6]. An automated algorithm for classifying CAD images is highly necessary for nuclear physicians since the increasing number of cases causes a bottleneck for the doctors [7]. Concerning CAD diagnosis in nuclear image analysis, ML has been introduced and investigated by various research studies conducted so far, as a methodology for automatic classification [1,8,9,10,11,12]. Already, CAD systems have proven themselves as a highly stable method in the domain of cardiovascular data analysis, due to their ability to extract data in highly respected analyses, such as SPECT MPI. However, CAD systems can work together with ML techniques to provide automatic SPECT image classification, without the need for additional data. DL methodologies and especially convolutional neural networks (CNNs) have exhibited promising results in CAD diagnosis. Considering the diagnosis of imaging CAD, a wide number of studies have focused on the development of CNNs, as they exhibit high reliability in image classification. In what follows, the most prominent and relevant CNNs along with various efficient architectures previously proposed to classify SPECT MPI scans for the detection of CAD, are demonstrated, according to the reported research papers.

In particular, Berkaya et al. in [4] sought to correctly classify SPECT MPI images and diagnose any myocardial abnormalities such as ischemia and/or infarction. To do so, the authors developed two classification models, one DL-based model, and one knowledge-based model. The proposed models extracted results such as an accuracy of 94%, a sensitivity of 88%, and a specificity of 100%, which are close to those based on expert analysis, so they can assist in clinical decisions regarding CAD. Papandrianos et al. [13] explored the abilities of CNNs to automatically classify SPECT MPI images, in a two-class classification problem, where the possible outputs were normal and ischemia. The outcomes (AUC 93.77% and accuracy 90.2075%)_indicate that the proposed model is a promising asset in nuclear cardiology. Furthermore, Papandrianos et al. [8] dealt with the same problem, investigating the capabilities of the Red Green Blue (RGB)-CNN model while they further compared its performance with transfer learning. The proposed RGB-CNN emerged as an efficient, robust, and straightforward deep neural network (NN) able to detect perfusion abnormalities related to myocardial ischemia on SPECT images in nuclear medicine image analysis. Additionally, it was proven that this is a model of low complexity and generalization capabilities compared to state-of-the-art deep NNs. Betancur et al. in [14] combined semi-upright and supine polar maps to analyze SPECT MPI abnormalities. A CNN model was implemented and competed against the standard method of total perfusion deficit (TPD) towards the prediction of obstructive disease. The authors concluded that the DL approach extracted promising results (Area under Curve (AUC) per vessel 0.81, AUC per patient 0.77), in contrast to quantitative methods. Moreover, in another research study, Betancur et al. [15] explored deep CNNs utilizing MPI images to automatically predict obstructive disease. Raw and quantitative polar maps were used in stress mode. Although the results seemed adequate (AUC per vessel 0.76, AUC per patient 82.3%), further investigations are needed regarding this task.

Other studies on CAD diagnosis in nuclear medicine are focused on the implementation of the DL methods to categorize polar maps into normal and abnormal. Liu et al. [16] developed a DL model utilizing CNN to diagnose automatically myocardial perfusion abnormalities in stress MPI count profile maps. The CNN model delivers an output being characterized as normal or abnormal. The DL method was compared against the standard quantitative perfusion defect size method. This study concludes that DL for stress MPI has the potential in providing notable help in clinical cases. Zahiri et al. in [3] suggested the implementation of a CNN to classify SPECT images for a successful CAD diagnosis. The dataset consists of polar maps demonstrated in stress and rest mode. The extracted results concerning AUC (0.7562), sensitivity (0.7856), and specificity (0.7434), represent the ability of the network to assist in future applications with regard to SPECT MPI. Apostolopoulos et al. in [17] investigated the implementation of a CNN to automatically characterize polar maps that were acquired with the MPI procedure. The aim of this research was the diagnosis of CAD by using both attenuation-corrected and non-corrected images. According to its evaluation results (74.53% accuracy, 75% sensitivity, and 73.43% specificity), the DL model performed surprisingly well.

Apostolopoulos et al. in [18] developed a hybrid CNN-random forest (RF) model and utilized polar map images and clinical data for the classification of patients’ status into normal and abnormal and further compared the results against nuclear experts’ diagnosis. The proposed model was evaluated with the 10-fold cross-validation technique and achieved 79.15% accuracy, 77.36% sensitivity, and 79.25% specificity while exhibiting similar overall results to those provided by experts. Nakajima et al. in [19] developed an ANN for the diagnosis of CAD, in contrast to the conventional quantitative approach. The proposed model classified 364 external data for validation purposes. The applied ANN outperformed with 0.92% AUC. Filho et al. in [20] compared four ensemble ML algorithms, namely: adaptive boosting (AB), gradient boosting (GB), eXtreme gradient boosting (XGB), and RF for the classification of SPECT images into normal and abnormal. The proposed model was evaluated with the utilization of the cross-validation approach and achieved an AUC of 0.853%, accuracy of 0.938%, and sensitivity of 0.963%. Ciecholewski et al. in [21] presented three methodologies: SVM, PCA (principal component algorithm) and NN, to diagnose ischemic heart disease. The results showed that SVM achieved greater accuracy (92.31%) and specificity (98%), whereas PCA extracted the best sensitivity.

Additionally, various recent studies demonstrate outstanding achievements regarding explainable artificial intelligence (XAI) implementations, which manage to provide transparency in the case of black-box models [22]. In particular, utilizing GRAD-CAM for the diagnosis of CAD, Otaki et al. [23] developed an explainable DL model, which highlights the regions that correspond to the predicted outcome. To evaluate the model, the authors compared its outcome against those provided by the automated quantitative total perfusion deficit (TPD) and the experts’ diagnosis. The results demonstrated that the DL model outperformed all. Spier et al. in [24] explored the ability of graph convolutional neural networks (GCNNs) to diagnose CAD successfully. GCNNs employ the same operation as that of CNNs to maps without the need for re-sampling them. The authors developed a model that can provide a second opinion on the diagnosis of CAD by acquiring polar maps. Chen et al. in [1] explored 3D gray CZT SPECT images and focused their research on the development of a CNN and the application of Grad-CAM capable of successfully diagnosing CAD. The model achieved an accuracy of 87.64%, sensitivity of 81.58%, and specificity of 92.16%. Otaki et al. in [25] developed a CNN model for the classification of polar maps into normal and abnormal, in contrast with TPD. Clinical data were added, such as sex and BMI. Additionally, Grad-CAM was applied to visualize the regions that correlate to the predicted output. The sensitivity produced by the CNN approach was 82% and 71% for women in the dataset, while the TPD for upright and supine was 75% and 73% in men and 71% and 70% in women, accordingly. It is worth mentioning that XAI models have not been examined thoroughly in nuclear medicine; thus, it is a promising and open research field in CAD diagnosis using SPECT MPI images.

According to previously conducted studies and the corresponding results, CNNs seem to attain similar accuracy to that of medical experts, and the approaches developed [8,10,11]. However, it emerges that there are not any studies performing the three-class categorization into ischemia, infarction, and normal. On that basis, the aim of this research work focuses on information extraction from SPECT data, classified by an expert reader, towards an automatic classification of obstructive CAD images, with three possible outputs: infarction, ischemic, and normal. The innovation involves the contribution of a predefined RGB-CNN architecture, which has been concluded after a thorough exploration of parameters, and the proposal of pre-trained models, which are VGG-16 and DenseNet-121, for improving CAD detection. The evaluation of the model is accomplished through reliable metrics and the usage of k-fold cross-validation.

The contribution of this research is to propose a CNN model and utilize transfer learning with the purpose of developing a model that can automatically classify SPECT images and predict CAD. For that reason, we have carefully selected metrics, that are reliable and robust such as accuracy, loss, AUC value, and ROC curve [4]. Our extracted results demonstrate high constancy and so our produced models can assist in future nuclear imaging issues. The structure of this article is as follows: we first present a compact yet detailed literature review on the research community’s contributions to DL and ML processing of SPECT MPI imaging exploring the notions of transfer learning, clustering, and classification. Following, the materials and a small description of the available datasets are discussed. In the next section, we illustrate the methodology of processing in a predefined sequence of steps thus, prototyping a directed process imposed by the CNN implementation, validation, and testing. Finally, we provide an in-depth presentation of the outcomes and a discussion of the results of the tri-partite classification of the available images according to CAD conditions.

## 2. Materials and Methods

### 2.1. CAD Dataset

Patient data have been acquired from the Diagnostic Medical Center “Diagnostiko-Iatriki A.E.” in Larisa, Greece by the Nuclear Department and have been retrospectively examined. The study covers a period from 30 March 2012 to 28 February 2017. Over this period, 842 consecutive patients underwent gated-SPECT MPI with 99mTc-tetrofosmin. A hybrid SPECT/CT gamma-camera system (Infinia, Hawkey-4, GE Healthcare, Chicago, IL, USA) was used for MPI imaging. Fifty-six (56) patients were excluded from the dataset due to inconclusive MPI results. Our dataset includes a total of 647 patients, where 262 of them correspond 134 to infarction, 251 to ischemic, and 262 to normal. The images have been extracted with SPECT method and are a visual representation of the heart in rest and stress mode.

Two nuclear medicine experts (N. Papandrianos and N. Papathanasiou) were asked to label each instance of the dataset according to their expertise and experience. Both count several years of experience. Labeling had been performed by the experts using solely the MPI scans of each patient. In this way, the model could be directly compared with the human experts. Hence, this study uses the experts’ diagnostic yield as the ground truth and aims to furnish a DL model capable of competing with the human eye and the human expertise. The Ethical Committee approved the study of our institution. The nature of the survey waives the requirement to obtain patients’ informed consent. In Figure 1 we have a representation of all cases, original SPECT MPI scans, which clearly present the stress and rest mode for each one of the three axes, short-axis (SA), vertical long-axis (VLA), and horizontal long-axis (HLA).

In the case of infarction, as depicted in Figure 1a, the arrows indicate the fixed defects due to scar or infarct, being represented as large hypoperfused segments at post-stress and at rest. As regards Figure 1b which involves the ischemia case, the arrows indicate the segments with stress-induced hypoperfusion (flow heterogeneity) against normal perfusion at rest. Regarding the variation of colors in MPI scans, the color bar in the far right of rows A and B, displays the myocardial perfusion range from normal (yellow) to low (deep blue and black).

### 2.2. Research Methodology

#### 2.2.1. Convolutional Neural Networks: Main Aspects

CNNs have become an encouraging type of model for medical image analysis since they do not need manual extraction of features, and instead, they can acquire hierarchical features by processing low-level features into high ones [26].

The first layer in the implementation of CNN is the convolutional layer, which is the core function, where the feature maps are generated. The feature maps are the result of the application of filters to the input image, for the purpose of attaining the pixels that have a positive impact on the output.

Following is the pooling layer, which reduces the spatial size by forwarding the necessary data for the optimization of calculation time and the usage of minimum computed resources, during the training. Next, a dropout layer is included, which randomly nullifies neurons on each iteration, to prevent overfitting. Then, flatten layer follows, which transforms multidimensional data into a one-dimension array, for inserting it into a fully connected layer. Lastly, a fully connected layer follows, where each node is connected with the previous node for the calculation of output, with the utilization of the activation function.

Concerning the activation functions, we utilize RELU (rectified linear unit) for all convolutional and fully connected layers, which outputs only the values that are positive and nullifies all negative values. The activation function of the output layer is based on the output, where in binary classification problems, the sigmoid function is utilized, where it contains a threshold, and any value higher than the threshold are classified as positive and the rest as negative. In a multiclass classification problem, the softmax function is preferred, which outputs a probability value for each possible label, and the higher probability indicates the predicted label [8].

#### 2.2.2. Methodological Framework

This paper targets the implementation of an RGB-CNN model, as a prosperous method in nuclear imaging for the classification of SPECT data and utilization of transfer learning. The robust CNN that was applied in our dataset can export highly efficient results. In Figure 2 the main methodological framework of the proposed approach is presented.

The steps of our methodology are reported below:

Step 1: Loading dataset

The SPECT MPI images were given by a nuclear physician, for the construction of an automatic classification tool. The images are demonstrated in RGB mode, and the corresponding patients underwent stress and rest examination. The dataset includes images of three categories: infarction, ischemia, and normal, and they are labeled “0”, “1”, and “2” respectively.

Step 2: Data preparation

Data normalization: Normalizing data is a common technique in the DL methodology. It is important to rescale the values of pixels and transform them in the range of (0, 1), so there would be an easier computation process.Data shuffle: To achieve generalizability and avoid bias in our proposed CNN model, it is mandatory to shuffle our data, so the data will be inserted into the model in random order.Data split: The dataset was split into three parts: training, validation, and testing. Each part has a specific percentage of the whole dataset, more specifically, training holds 85% of the total dataset and testing holds the remaining 15%. From the training dataset, 80% is provided for training, whereas 20% of training is supplied for validation [13].

Step 3: Training

Data augmentation: The data augmentation technique is important in DL applications, to increase the number of the existing dataset, especially when the number of training data is small. The techniques that are applied in the current research are: rescale, rotation, range, width shift range, height shift range, shear range, zoom range, horizontal flip, and vertical flip. The aim of this procedure is to achieve generalizability and prevent overfitting, which occurs when the model fits precisely with training data and cannot perform correctly with unseen data [17,24].Define CNN architecture and activation functions: For the definition of the best-proposed model, which assures generalizability and robustness, a thorough exploration must be made. The parameters that need to be explored are pixel size, batch size, number of nodes and layers of convolutional layers, and number of nodes of fully connected layers. Furthermore, it is essential to select the proper activation function for the convolutional and dense layers [24].Train CNN: During the training process, CNN extracts abstract patterns, which are acquired from the produced feature maps with respect to the corresponding output. Furthermore, the back-propagation technique is utilized for the fine-tuning of the model with the adjustment of weights, that have a positive impact on the result, and nullifying the values that do not [4,27].

Step 4: Validation

Throughout the process of validation, the network has not been fully specified and utilizes a section of training data to extract validation metrics. The purpose of the validation is to minimize the errors, normalize the weights and provide the best results.

Step 5: Testing

During the testing procedure, the best model has been concluded. The goal of testing is to evaluate the results of the model in classifying unknown data and for that reason, we used valid metrics such as precision, recall, sensitivity, specificity, accuracy, loss, and AUC [8,16,27].

#### 2.2.3. The Proposed CNN Architecture

The aim of this research is the development of an efficient CNN that can fully autonomously identify any signs of cardiovascular disease. With the utilization of accurate metrics along with external evaluation and more specifically, 10-fold cross-validation the CNN model will be tested for its accuracy and robustness. Throughout the CNN exploration process, experiments were conducted with various values for the following parameters: pixel size, batch size, number of nodes and layers of convolutional layers, and number of nodes of fully connected layers [27]. The proposed structure for the corresponding dataset is illustrated in Figure 3 and includes four convolutional layers, followed by one pooling layer and one dropout layer, whereas two fully connected layers follow which both have 128 nodes, providing three outputs: infarction, ischemic, and normal. Regarding the activation function of the output layer, we used softmax, because it is suitable for multi-class classification problems [27].

## 3. Results

In this research study, the problem we seek to address is the automatic classification of obstructive CAD images, with three possible outputs: infarction, ischemic, and normal. To that end, various architectures of CNN, which are commonly used for classifying images correctly [24,26,27,28,29], were explored to find the ideal for our corresponding dataset, while ensuring stability and generalizability. Each combination of the examined architectures was performed at least 10 times, to confirm its reliability.

The developed algorithm was run in Google Colab [21,30] runtime environment, which gives access to a powerful machine with faster GPUs and an increased amount of RAM and disk, making it suitable for training large-scale ML and DL models. This platform provides free space for uploading data, allowing users to run code entirely on the cloud and thus, overcoming any computational limitations on local machines. With reference to the development of our code, concerning the proposed CNN, we used Python 3.6 version, utilizing Tensorflow 2.6 and Keras 2.4.0. framework packages. The OpenCV library was used to load our dataset, while the scikit-learn library was used for splitting our dataset and computing the results. Matplotlib was further deployed for representing the plots. Furthermore, the specifications of our PC for the main component are: (1) Processor: Intel^®^ Core™ i9-9980HK CPU 2.40GHz, (2) GPU: NVIDIA Quadro RTX 3000, (3) Ram: 32 GB, (4) System operating type: 64-bit operating system and x64-based processor.

To find the best architecture for the provided dataset, a thorough exploration was conducted. On that basis, we examined different values for pixel sizes (200 × 200, 250 × 250, 300 × 300 and 350 × 350), batch sizes (8, 16, 32 and 64), number of nodes for convolutional layers (32-64-128, 16-32-64-128 and 16-32-64-128-256) and nodes for dense layers (32-32, 64-64, 128-128 and 256-256). Additionally, we examined various epochs (400 up to 800), whereas we kept a fixed drop rate value of 0.2 for all simulations, as this value provides adequate results.

To properly assess the results, certain robust metrics were selected, such as accuracy, loss, AUC, ROC curve, precision, recall, sensitivity, and specificity. More specifically, accuracy is the ratio between several correct predictions to a total number of predictions. Loss dictates the error in classifying images by calculating the gradients between predicted and real outputs. The AUC value is the numerical demonstration of a model to differentiate an image within the possible output and its value ranges between 0 and 1, where 1 is the ideal, ROC curve is the visual representation of the classifier’s performance. Regarding precision is the ratio between correctly classified positive samples to the total number labeled as positive, where recall is the percentage of correctly classified samples. Sensitivity evaluates the model’s capability by calculating the percentage of the positive samples, that were predicted correctly, whereas specificity computes the percentage of negative samples [8,11,27].

With respect to the initial default values for the performed exploration process, these are the following: 250 × 250 for pixel size, 16 for batch size, 16-32-64-128 for convolutional layers, and 128-128 for dense layers, as well as 400 epochs. Firstly, various batch sizes were examined by keeping the rest of the values fixed, to find the best batch size. Table 1 presents the respective results, for each of the examined batch sizes. It emerges that for batch size 32 the overall results are superior to those of the rest batch sizes.

Afterwards, we examined different pixel sizes, while having kept fixed values for convolutional and dense layers, considering that 32 is the best batch size. In Figure 4, we can clearly distinguish, the remarkable results that 300 × 300 extracted, in contrast to the rest structures, thus being the best combination for our corresponding dataset.

In the following step, different convolutional layers were examined, in order to configure to the best combination. In Figure 5 the calculated results are visually illustrated regarding the application of various convolutional layers. The combination of 16-32-64-128 convolutional layers seems to have performed better, in terms of overall metrics. On the other hand, the combination 32-64-128 and 16-32-64-128-256 did not perform sufficiently. Moreover, we have performed further experiments with various epochs, in the range between 400 and 800. However, we observed that above 400 epochs, no improvement in results was attained.

The last step deals with the selection of the best number of nodes for the dense layers. Therefore, different numbers of nodes were examined, and the produced results are visually depicted in Figure 6. It is concluded that the best sequence is 128-128, where all metrics are considered.

At this point, various architectures were explored for our RGB model, and each combination was executed for at least 10 runs so that a robust and reliable model was finally conducted. The ultimate architecture of RGB is 300 × 300 for pixel size, 32 for batch size, 16-32-64-128 for convolutional layers, and 128-128 for dense layers. We can see in Table 2, the average value of metrics for each class individually. It is obvious that the model can perfectly distinguish each class.

In addition to the procedure, which is followed for the determination of the RGB model, we also utilized a powerful method, commonly used in imaging problems, which is called transfer learning. A model is previously trained in various classification problems and can perform adequately in related tasks and has great generalizability skills [4]. In our proposed research, we used two pre-trained models, Visual Geometry Group (VGG)-16 and DenseNet-121. Furthermore, we selected the architecture for the pre-trained networks, following the literature on VGG-16 and DenseNet-121 applications in medical image analysis, as well as our previous works in this domain. Regarding VGG-16, the best architecture entails 200 × 200-pixel size, 32 batch size, 0.2 drop rate, 400 epochs, 14 true trainable parameters, global average pooling, and two fully connected layers with 32 nodes each. It must be understood that VGG-16 has 16 trainable layers, as the name indicates, and for our comparative process, we “froze” the last two layers of VGG-16, which means that the weights of these layers will not be updated during the training procedure to include two fully connected layers, and to adjust the computation of output for our corresponding dataset. In Table 3, we can see how well each class of the model was distinguished by visualizing the result of robust metrics.

DenseNet-121’s best architecture consists of 250 × 250 for pixel size, 32 for batch size, 0.2 drop rate, 400 epochs, 12 true trainable layers, global average pooling, and 2 fully connected layers with 32 nodes each. In Table 4, the results of the metrics are demonstrated, and they represent the efficiency of DenseNet-121 to distinguish classes.

In Table 5, we have the comparison of results among the RGB and the pre-trained networks (VGG-16, DenseNet-121). We can see that RGB performed efficiently, where VGG-16 extracted adequate metrics and DenseNet-121 extracted the lowest results. Regarding computation time, RGB-VGG-16 are similar and DenseNet-121 has the highest calculation time.

In Figure 7, Figure 8 and Figure 9 the accuracy and loss plots are demonstrated for each model. RGB performed better. In Figure 10, Figure 11 and Figure 12, we can see the ROC curves for each model and each class individually. It is clear that RGB performed better.

In Figure 13, the comparison of RGB-CNN, VGG-16, and DenseNet-121 is illustrated with respect to the produced validation and testing accuracy, AUC, and validation and testing loss results for each model. It is observed that RGB extracted the best results, for the case where VGG-16 has similar results in validation and testing loss. DenseNet-121 performed adequately.

To implement robust and reliable models, we utilized the 10-fold cross-validation technique in order to further evaluate the performance of our model [6]. This method splits the given dataset into ten parts using every time, nine parts for training, and one part for testing. This procedure is repeated until every chunk is utilized as a testing part. In Table 6, Table 7 and Table 8 we have demonstrated the results for RGB, VGG-16, and DenseNet-121, accordingly, whereas in Table 9, Table 10 and Table 11 the performance metrics of each CNN model for each possible output are depicted. RGB outperformed in contrast to pre-trained networks, which demonstrates great robustness and efficiency.

For comparative analysis, Table 12 summarizes the results of each one of the best CNN models applying 10-fold cross-validation. According to the produced results, RGB exhibits sufficient performance with robust results, VGG-16 performed efficiently, whereas DenseNet-121 extracted adequate outcomes.

## 4. Discussions

In this study, three CNN models were examined to differentiate between infarction, ischemic and normal perfusion in SPECT-MPI images; in stress and rest mode. The first DL-based model, the RGB-CNN, was built from scratch, comprising four convolutional layers and two dense layers. VGG-16 was the second model applied, which is acknowledged for image classification tasks and was designed using fourteen layers, to prevent overfitting. The third model is DenseNet-121, which was implemented with twelve layers for the same purpose. Transfer learning was applied to the last two models, as a widely used and robust technique for image classification. The provided dataset consists of 647 images, where the data are heterogeneous, which is an important factor in terms of generalizability. Since the dataset is considered rather small, the data augmentation method was employed to increase the size. Furthermore, the dataset was split into three parts, training, validation, and testing, to evaluate our model in data, with known outputs, through the validation dataset, and unseen data, with the utilization of the testing dataset. Additionally, a 10-fold cross-validation was performed, and the results demonstrated great stability and reliability for our models.

Nuclear experts also performed visual classification of the models achieving supervised learning with known output. It should be highlighted that we conducted an in-depth exploration process where various combinations concerning different image sizes, batch sizes, and convolutional and dense layers were examined, to conclude the best architecture of the DL models, for the corresponding dataset. Concerning the selection of batch size, we investigated different values for batch sizes between 8 and 64, following the reported literature [31,32]. The overall outcomes of the research pinpoint that a small number (16, 32) of batch sizes can offer reliable results. After a thorough exploration process, it emerged that batch size 32 provides the best results for CAD diagnosis demonstrating a better training process with a low learning rate, thus avoiding overfitting and performing with higher classification accuracy. It can be definitely concluded that the research findings reflect the results of this study.

The distinction of results and the improvement of metrics, during the experiment, is a promising fact for the model’s capabilities. In this direction, certain metrics, which are robust and reliable, were employed to evaluate models’ stability and performance. These are accuracy, loss, AUC, ROC curve, precision, recall, sensitivity, and specificity and were applied in all three models. The results demonstrated that all three models can achieve high accuracy and minimize loss, even for a small dataset, which is a key factor for attaining reliability. VGG-16 and RGB-CNN performed better against DenseNet-121, achieving an accuracy of 93.33%, 88.54%, and 86.11%, respectively. These results ensure reliability and that fine-tuning a model is both important and effective. It turns out that the proposed RGB-CNN model and other robust and widely recognizable models can perform equally well.

Due to their generalizability and reliability, as proven through k-fold cross-validation, the proposed models can be deployed by nuclear experts on any SPECT-MPI images, without any further adjustments, extracting sufficient results. However, there are some limitations in this work, that need to be considered and further addressed in the future. Firstly, only SPECT images were used as input to the proposed models, since CNNs can perform directly to images. Secondly, the size of our dataset is limited for CNN classifiers, so we had to increase it by adding more images to feed our models, thus extracting more abstract feature maps to further optimize our approach. Despite the existence of these limitations, the proposed models are capable to perform satisfactorily and operate adequately in unknown data.

Some of the clinical implications that emerge from the employment of the proposed methodology in medical imaging include the invaluable support towards an instant and automatic clinical diagnosis of SPECT MPI images that could prevent patients from possible undesirable heart conditions, such as infarction. The DL-based approach could potentially serve as a crucial assisting tool for medical experts in their effort to (i) evaluate and further diagnose correctly SPECT MPI images of patients suffering from angina, or known coronary artery disease, and (ii) instantly deliver proper treatment suggestions. The integration of the proposed RGB-CNN model in the clinical everyday practice constitutes a challenge in the medical imaging domain, provided that its enhanced performance is also validated on larger datasets.

A notable aspect that has been reported in the present studies is explainable artificial intelligence (XAI). Even though CNNs can extract spectacular results, it remains unclear how CNN models arrive at a specific output resulting in certain decisions. CNNs are considered black boxes, as they do not provide any evidence of their conclusions and thus, we depend mostly on the extracted metrics to fully trust their output. CNNs accept images as input, compute some calculations with the help of hidden layers, and produce an output in the form of a percentage of a prediction. Therefore, scientists and medical experts have the belief that CNNs perform with bias and therefore, they are skeptical about CNNs trustworthiness, especially in the medical domain, where each decision is crucial for the patient’s health. Considering this persistent disbelief of their abilities and trustworthiness, explainable artificial systems have been developed in order to provide interpretability and understanding of the internal computation of CNNs. In future studies, our aim is to thoroughly explore and further develop XAI systems. In view of the abovementioned considerations, XAI is a realm that needs to be explored to a greater extent in the future, especially for medical diagnosis, where the predicted output of the proposed models has such an essential role [33].

To sum up, our three-class classification models are capable of performing pleasingly and distinguishing images between infarction, ischemic and normal cases, with the utilization of 10-fold cross-validation, transfer learning, and data augmentation. All three models can be an integral tool to assist with the automatic diagnosis of coronary artery disease and can be clinically deployed with the help of a user interface (UI) platform.

## 5. Conclusions

DL has demonstrated that it can perform autonomously and assist in the healthcare domain [34,35]. More specifically, CNNs are benefited from their ability to accept images as input and achieve high accuracy in image classification. In the current study, an RGB model was proposed, implemented from scratch, and two models VGG-16 and DenseNet-121 were utilized by applying the transfer learning method. The goal of this study was to address the three-class classification problem regarding SPECT-MPI images to diagnose CAD providing possible outputs as infarction, ischemia, and normal. Because of the small dataset, data augmentation was employed to increase the size of the dataset, and a 10-fold cross-validation was performed to further evaluate the proposed model. An exploratory analysis was conducted to discover the best architecture for all three models regarding the dataset, and to achieve generalizability, as well. All the proposed models extracted promising results, despite the small dataset, such as high accuracy, minimum loss, and tremendous value for AUC, precision, recall, sensitivity, and specificity, exhibiting the ability to offer reliable and valuable help to nuclear experts. In future studies, we plan to develop XAI to describe CNNs, so there could be transparency in decision-making concerning image prediction while observing if there is a bias behind CNN’s decision to classify an image.

## Figures and Tables

**Figure 1 jcm-11-03918-f001:**
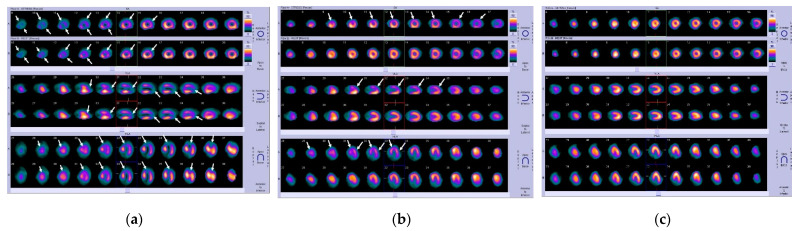
Original SPECT MPI samples of coronary artery disease produced by Siemens camera: (**a**) infarction; (**b**) ischemia; (**c**) normal.

**Figure 2 jcm-11-03918-f002:**
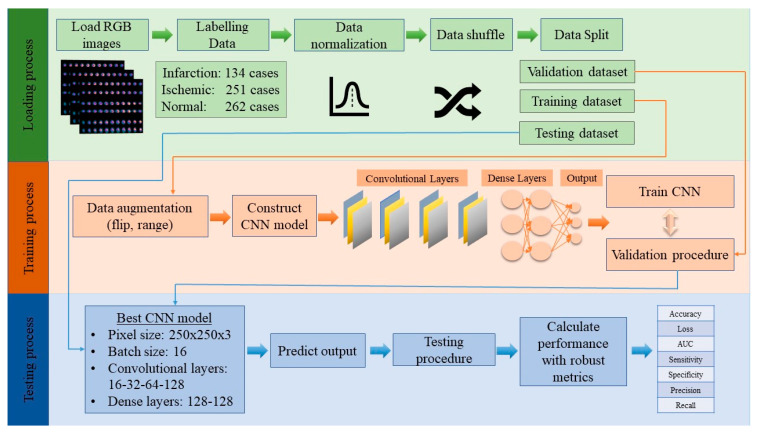
Steps of the proposed methodology.

**Figure 3 jcm-11-03918-f003:**
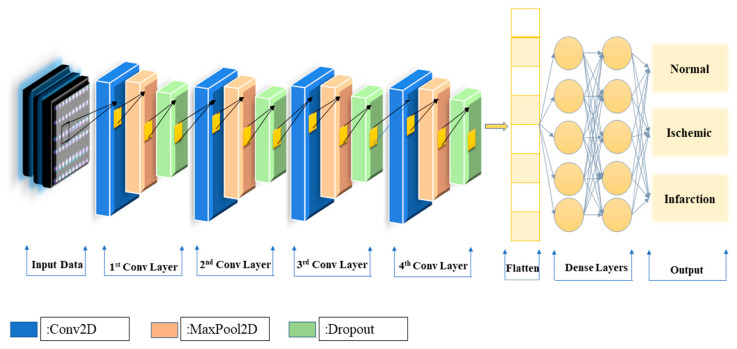
The CNN framework for coronary artery classification.

**Figure 4 jcm-11-03918-f004:**
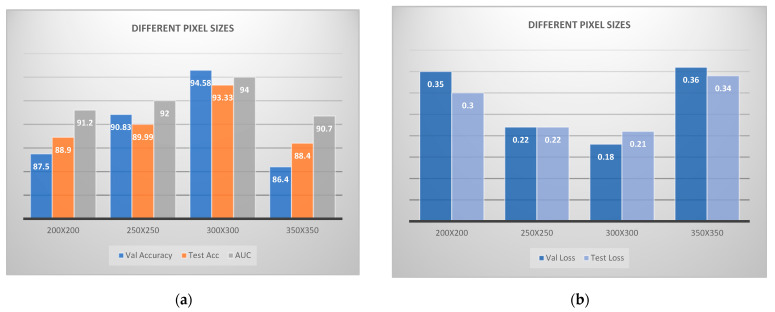
Convolutional neural network architecture for CAD classification problem comparing various pixel sizes: (**a**) accuracies; (**b**) losses.

**Figure 5 jcm-11-03918-f005:**
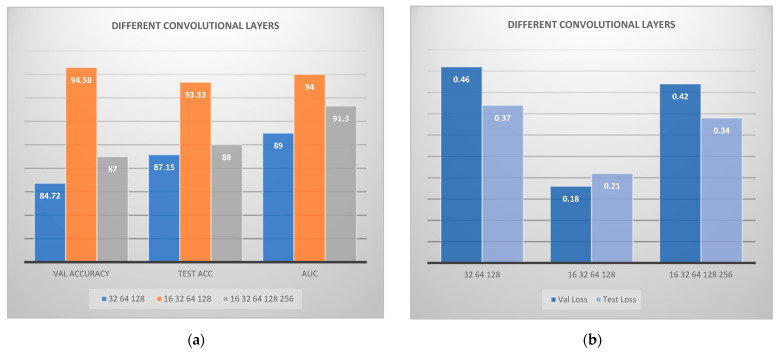
Convolutional neural network architecture for CAD classification problem comparing various convolutional layers: (**a**) accuracies; (**b**) losses. VAL is the validation accuracy and ACC is the accuracy.

**Figure 6 jcm-11-03918-f006:**
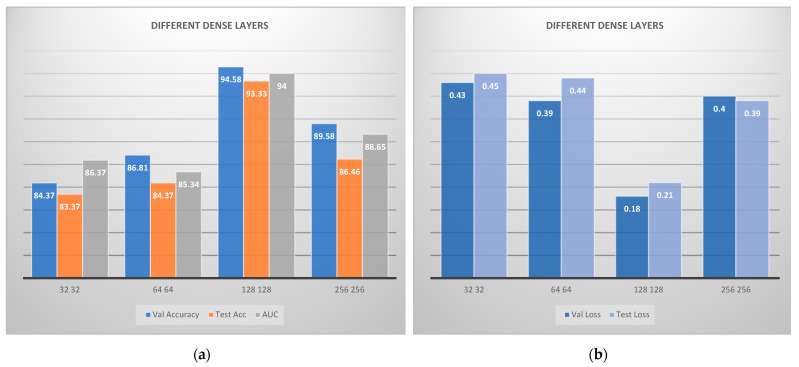
Convolutional neural network architecture for CAD classification problem comparing various fully connected layers: (**a**) accuracies; (**b**) losses. VAL is the validation accuracy and ACC is the accuracy.

**Figure 7 jcm-11-03918-f007:**
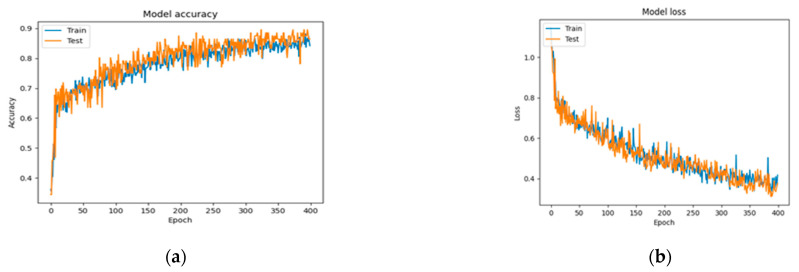
Plots demonstrating the best RGB-CNN for the CAD classification problem: (**a**) accuracy; (**b**) loss.

**Figure 8 jcm-11-03918-f008:**
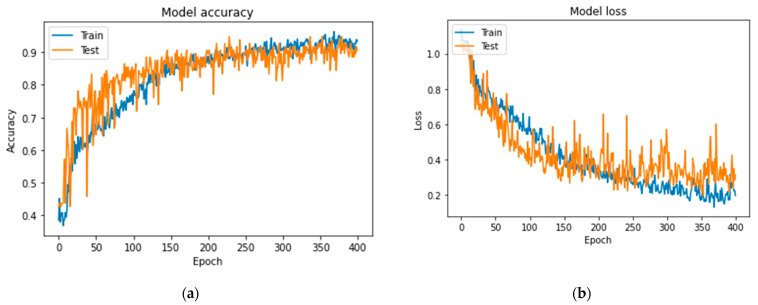
Plots demonstrating the VGG-16 for comparing reasons, for the CAD classification problem: (**a**) accuracy; (**b**) loss.

**Figure 9 jcm-11-03918-f009:**
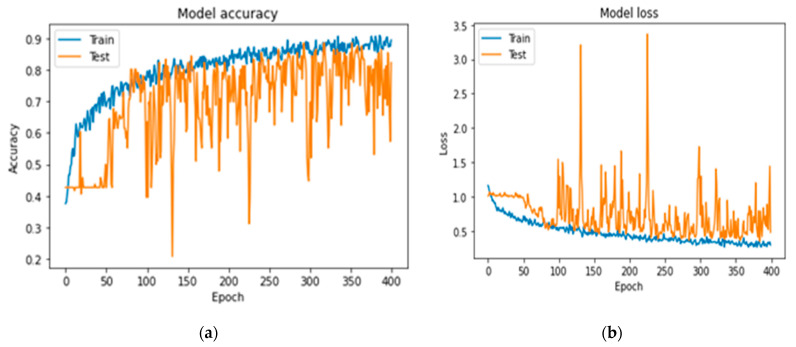
Plots demonstrating the DenseNet-121 for comparing reasons, for the CAD classification problem: (**a**) accuracy; (**b**) loss.

**Figure 10 jcm-11-03918-f010:**
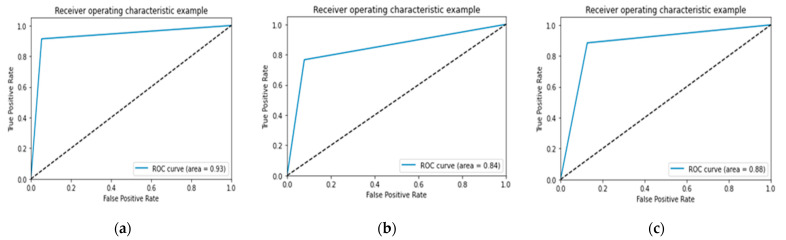
Performance of Receive Operating Characteristic (ROC) curve for proposed RGB: (**a**) infarction; (**b**) ischemia; (**c**) normal.

**Figure 11 jcm-11-03918-f011:**
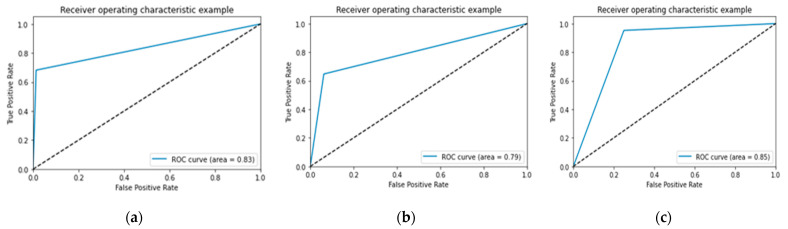
Performance of ROC curve for VGG-16: (**a**) infarction; (**b**) ischemia; (**c**) normal.

**Figure 12 jcm-11-03918-f012:**
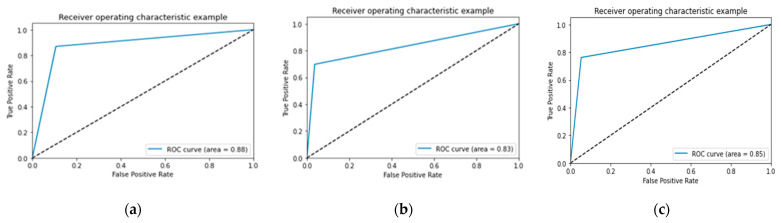
Performance of ROC curve for DenseNet-121: (**a**) infarction; (**b**) ischemia; (**c**) normal.

**Figure 13 jcm-11-03918-f013:**
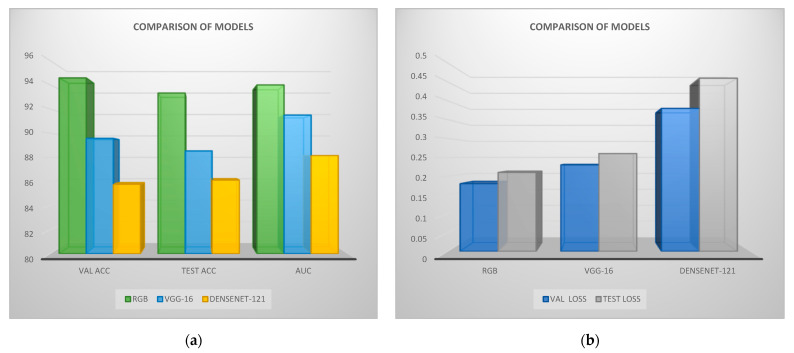
Performance of validation and testing metrics and AUC of Red Green Blue (RGB) and transfer learning: (**a**) accuracies; (**b**) losses. ACC is the accuracy and VAL the validation.

**Table 1 jcm-11-03918-t001:** Comparison of results of different batch sizes.

	Val Accuracy	Val Loss	Test Accuracy	Test Loss	AUC
8	87.7	0.32	84.58	0.4	0.86
16	88.54	0.39	90.62	0.26	0.92
**32**	**94.58**	**0.18**	**93.33**	**0.21**	**0.94**
64	84.76	0.4	82.81	0.54	0.87

**Table 2 jcm-11-03918-t002:** Presentation of values of efficient metrics of best RGB-CNN model for each class.

	Precision	Recall	Sensitivity	Specificity
Infarction	0.94	0.89	0.9	0.99
Ischemia	0.92	0.9	0.93	0.96
Normal	0.94	0.97	0.97	0.95

**Table 3 jcm-11-03918-t003:** Presentation of values of efficient metrics of the VGG-16 model for each class.

	Precision	Recall	Sensitivity	Specificity
Infarction	0.81	0.98	0.81	0.99
Ischemia	0.96	0.86	0.96	0.93
Normal	0.97	0.95	0.96	0.96

**Table 4 jcm-11-03918-t004:** Presentation of values of efficient metrics of DenseNet-121 model for each class.

	Precision	Recall	Sensitivity	Specificity
Infarction	0.97	0.55	0.62	0.99
Ischemia	0.92	0.15	0.91	0.8
Normal	0.93	0.95	0.91	0.93

**Table 5 jcm-11-03918-t005:** Comparison of efficient metrics, between RGB and Transfer Learning Methods.

	Validation Accuracy	Validation Loss	Testing Accuracy	Testing Loss	AUC	TIME
**RGB**	94.58	0.18	93.33	0.21	0.94	**3256 s**
VGG-16	89.58	0.23	88.54	0.26	0.91	3250 s
DENSENET-121	85.76	0.38	86.11	0.46	0.88	3593 s

**Table 6 jcm-11-03918-t006:** Runs of 10-fold cross-validation for the proposed Red Green Blue (RGB).

Folds	Accuracy	Loss
Fold 1	92.59	0.22
Fold 2	90.74	0.28
Fold 3	92.6	0.21
Fold 4	93.19	0.18
Fold 5	90.74	0.26
Fold 6	92.59	0.17
Fold 7	92.59	0.19
Fold 8	93.31	0.18
Fold 9	93.33	0.15
Fold 10	92.42	0.25
**Average:**	**92.4 (±2.24)**	**0.21**

**Table 7 jcm-11-03918-t007:** Runs of 10-fold cross-validation for the VGG-16.

Folds	Accuracy	Loss
Fold 1	89.09	0.31
Fold 2	86.36	0.49
Fold 3	89.99	0.32
Fold 4	87.72	0.47
Fold 5	87.27	0.32
Fold 6	90.9	0.28
Fold 7	88.18	0.27
Fold 8	86.36	0.3
Fold 9	90.9	0.24
Fold 10	89.09	0.26
**Average:**	**88.09 (±2.38)**	**0.33**

**Table 8 jcm-11-03918-t008:** Runs of 10-fold cross-validation for the proposed DenseNet-121.

Folds	Accuracy	Loss
Fold 1	89.99	0.29
Fold 2	86.36	0.37
Fold 3	89.09	0.3
Fold 4	85.45	0.46
Fold 5	87.27	0.32
Fold 6	88.18	0.39
Fold 7	84.54	0.35
Fold 8	89.09	0.36
Fold 9	75.45	0.55
Fold 10	75.45	0.64
**Average:**	**85.09 (±3.94)**	**0.4**

**Table 9 jcm-11-03918-t009:** Presentation of metrics of proposed RGB for each class, after a 10-fold procedure.

	Precision	Recall	Sensitivity	Specificity
Infarction	0.99	0.91	0.915	0.99
Ischemia	0.96	0.93	0.93	0.97
Normal	0.93	0.99	0.99	0.95

**Table 10 jcm-11-03918-t010:** Presentation of metrics of VGG-16 for each class, after a 10-fold procedure.

	Precision	Recall	Sensitivity	Specificity
Infarction	0.84	0.92	0.73	0.75
Ischemia	0.93	0.85	0.68	0.77
Normal	0.9	0.95	0.75	0.77

**Table 11 jcm-11-03918-t011:** Presentation of metrics of DenseNet-121 for each class, after a 10-fold procedure.

	Precision	Recall	Sensitivity	Specificity
Infarction	0.81	0.82	0.65	0.69
Ischemia	0.87	0.74	0.6	0.71
Normal	0.89	0.83	0.65	0.73

**Table 12 jcm-11-03918-t012:** Comparison between proposed RGB-CNN and pre-trained networks in testing metrics and AUC, following the application of default data split method and 10-fold.

	Data Split (80–20% Testing)			10-Fold		
	Testing Accuracy	Testing Loss	AUC	Testing Accuracy	Testing Loss	AUC
RGB-CNN	93.33	0.21	0.94	92.4	0.21	0.93
VGG-16	88.54	0.26	0.91	88.09	0.33	0.92
DenseNet-121	86.11	0.46	0.88	85.09	0.4	0.9

## Data Availability

The datasets analyzed during the current study are available from the nuclear medicine physician on reasonable request.

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
