# Peer review of "Deep Learning-Based Automated Diagnosis for Coronary Artery Disease Using SPECT-MPI Images"

_jcm, 2022, doi:10.3390/jcm11133918_

Round 1

Reviewer 1 Report

I have read this article with great interest and found very interesting. The authors developed three supervised learning-based artificial intelligence, which were RGB-CNN, VGG-16, and DenseNet-121, and tested their performance of classifying single photon emission computed tomography (SPECT) myocardial perfusion imaging (MPI) into three diagnostic categories: normal, ischemia, and myocardial infarction. Among them, the RGB-CNN model showed the highest accuracy and were followed by VGG-16 and DenseNet-121. The background is sound, the methodologies are shown in detail for other researchers to reproduce, and the results are understandable. However, I would suggest a couple of things that the authors might wish to address.

1. Although this article provides with sufficient information, some parts might be a bit redundant. For example, it might be better to combine the Introduction and the Literature sections together and make them more concise, or some content in the Literature can be moved to the Discussion section. For me, the Introduction and the Literature sections feel as if this article was a review article rather than an original research article.

2. The Materials and Methods sections might be a bit too technical, particularly for clinicians who are not familiar with artificial intelligence. Although the details of deep-learning structures, such as data preparation, convolutional layers, dense layers, activation functions, the number of epochs, etc., are important and informative for other researchers, they might not necessarily help readers to better understand the research, or might be overwhelming. I would suggest reducing the technical details as much as possible, and instead adding clinical implications that this research can provide.         

Author Response

RESPONSE REVIEWER#1

Thank you for your insightful comments. We have addressed them in the revised manuscript.

I have read this article with great interest and found very interesting. The authors developed three supervised learning-based artificial intelligence, which were RGB-CNN, VGG-16, and DenseNet-121, and tested their performance of classifying single photon emission computed tomography (SPECT) myocardial perfusion imaging (MPI) into three diagnostic categories: normal, ischemia, and myocardial infarction. Among them, the RGB-CNN model showed the highest accuracy and were followed by VGG-16 and DenseNet-121. The background is sound, the methodologies are shown in detail for other researchers to reproduce, and the results are understandable. However, I would suggest a couple of things that the authors might wish to address.

  1. Although this article provides with sufficient information, some parts might be a bit redundant. For example, it might be better to combine the Introduction and the Literature sections together and make them more concise, or some content in the Literature can be moved to the Discussion section. For me, the Introduction and the Literature sections feel as if this article was a review article rather than an original research article.

Response#1: We have properly addressed this comment by merging the Introduction and the Literature sections, making them shorter without altering the meaning in the revised manuscript. Table 1 has been deleted.

  1. The Materials and Methods sections might be a bit too technical, particularly for clinicians who are not familiar with artificial intelligence. Although the details of deep-learning structures, such as data preparation, convolutional layers, dense layers, activation functions, the number of epochs, etc., are important and informative for other researchers, they might not necessarily help readers to better understand the research, or might be overwhelming. I would suggest reducing the technical details as much as possible, and instead adding clinical implications that this research can provide.         

Response#2: We have properly addressed this comment by reducing the technical details as much as possible in the revised manuscript.

Moreover, the syntax of the text has been improved and errors have been corrected using an online proof-reading software.

Reviewer 2 Report

Dear Authors,
That was a nice contribution to the literature! The rather unusual structure of the manuscript (literature after the introduction and prior to the methods section ) makes it a little more difficult for the reader to grasp the whole paper. However, it is well-written, scientifically robust and provides novel insights for the relevant literature. Kindly replace the first sentences "In this research paper our main concern is Coronary Artery Disease (CAD), as it is the most usual type of heart disease; actually, heart disease is the primary reason for mortality, globally. The percentage of death attributed to CAD is higher, compared with other heart diseases.". It might be better to begin with:  "Recent evidence suggest that coronary artery disease (CAD) is the most common type of cardiovascular disease being the leading cause of mortality worldwide." 

Author Response

RESPONSE Reviewer#2

Thank you for your insightful comments. We have addressed them in the revised manuscript.

Dear Authors,
That was a nice contribution to the literature! The rather unusual structure of the manuscript (literature after the introduction and prior to the methods section ) makes it a little more difficult for the reader to grasp the whole paper. However, it is well-written, scientifically robust and provides novel insights for the relevant literature.

Kindly replace the first sentences "In this research paper our main concern is Coronary Artery Disease (CAD), as it is the most usual type of heart disease; actually, heart disease is the primary reason for mortality, globally. The percentage of death attributed to CAD is higher, compared with other heart diseases.". It might be better to begin with:  "Recent evidence suggest that coronary artery disease (CAD) is the most common type of cardiovascular disease being the leading cause of mortality worldwide." 

Response: We have addressed your main concerns regarding the structure of the manuscript. In particular, we have included the main literature into the Introduction section making the necessary modifications to make it more concise and easier to follow. Furthermore, we replaced the first sentences as per your suggestions which are much appreciated.

Moreover, the syntax of the text has been improved and errors have been corrected using an online proof-reading software.
